# Riddle of a Black Hole: Guessing the Riddle of a Black Hole

**Boris E. Meierovich**

P. L. Kapitza Institute for Physical Problems 2 Kosygina str., 119334 Moscow, Russia; meierovich@mail.ru

**Abstract:** A static structure of matter, extremely compressed to the state of a Bose–Einstein condensate by its own gravitational field, is considered. Instead of the widely spread restriction $\det g_{ik} < 0$, I used a weaker condition of regularity: all invariants of $g_{ik}$ are finite. This makes it possible to find regular static solutions to Einstein equations for a spherically symmetric distribution of matter with no restriction on total mass. In these regular static solutions, the metric component $g^{rr}$ changes its sign twice: $g^{rr}(r) = 0$ at $r = r_g$ and at $r = r_h > r_g$. The signature of the metric tensor is changed to $(+,+,-,-)$ within the spherical layer $r_g < r < r_h$. Though the gravitation dominates at extremely high density, I assume that it does not violate the exchange interaction of elementary particles of the Standard Model. The found regular static solution to Einstein equations, having no limitation on mass, pretends to describe the state of a black hole to which the gravitational collapse leads. The features of a collapsed black hole, its internal composition depending on total mass and the relation with surrounding dark matter, are considered. An astrophysical application: The pressure balance at the interface between a black hole and dark matter determines the plateau velocity of a galaxy rotation curve as a function of the black hole mass. The plateau velocity is inversely proportional to the black hole mass. The speed of rotation of a star at the periphery of a galaxy is proportional to the square root of the black hole mass (direct attraction to the center) and inversely proportional to the mass of the same black hole (as the influence of dark matter). For a condensate of massive bosons in the Standard Model, the direct attraction to the black hole and the influence of dark matter are equal if the black hole mass is about $\widetilde{M} \sim 4.24 \times 10^{37}$ g. In galaxies with black hole masses $M \gtrsim M_\odot = 1.989 \times 10^{33}$ g (like UMa: NGC 3726 and UMa: NGC 3769 of the Ursa Major cluster), the motion of stars is driven by dark matter. Their rotation curves should have a well-defined plateau. On the contrary, in galaxies with black hole masses $M >> \widetilde{M}$ (like in our Milky Way with the black hole mass $M = 8.6 \times 10^{39}$ g), the motion of stars is regulated by the black hole in the center. Dark matter does not play a significant role in our Milky Way Galaxy.

**Keywords:** black hole; dark matter; Bose–Einstein condensate

**PACS:** PACS numbers: 04.20.-q; 04.62.+v; 04.70.-s

## 1. Introduction

### 1.1. Einstein's Hypothesis

In the Schwarzschild static solution [1], there is a hypersphere $r = r_g$ where the components of the metric tensor $g_{00}$ and $g_{rr}$ "exchange signs". On this hypersurface, $g_{00}(r_g) = g^{rr}(r_g) = 0$. The vanishing of $g^{rr}$ means that the infinite small coordinate length in the radial direction corresponds to the finite "natural" length. This circumstance was not unnoticed by Einstein. Here is an excerpt from his article [2]: *"If it turns out that in some place of the four-dimensional continuum* $\det g_{ik}$ *vanishes, then it means that at this place the infinite small "natural" volume corresponds to the finite coordinate volume.* **We assume**

**that this is nowhere to be found**. *In this case,* det $g_{ik}$ *cannot change its sign; we will accept, in accordance with the special theory of relativity, that* det $g_{ik}$ **always has a finite and negative value.** *This assumption is a hypothesis about the physical nature of the considered continuum and, at the same time, a rule concerning the choice of a coordinate system."* (see [2], below Equation (18a)).

The point of this quote is that det $g_{ik}$ is negative provided that a singularity is nowhere to be found. Forgetting "**nowhere to be found**", people use the condition det $g_{ik} < 0$ unhesitatingly and everywhere. In the Schwarzschild metric, $g^{rr}$ is positive at $r < r_g$, and there is an inevitable singularity in the center $r = 0$. At a regular center, the ratio of the circumference to diameter tends to $\pi$ as $r \to 0$; wherein $g^{rr}(0) = -1$. The assumption "nowhere to be found" is violated in the center. For a real physical object of arbitrary high mass with a Schwarzschild gravitational field outside and a regular center, at least one more gravitational radius should exist inside the gravitating body.

The sign of the determinant det $g_{ik}$ is not an invariant. If det $g_{ik}$ changes sign on some hypersurface, while all invariants of $g_{ik}$ are finite, then this "singularity" is removable, and there exists a regular solution. The condition of regularity (all invariants of $g_{ik}$ are finite) is weaker than the requirement det $g_{ik} < 0$.

In the 1930s, in line with Einstein's hypothesis, the existence of a critical mass $M_{cr}$ for gravitating, spherically symmetric Fermi systems was established in the fundamental papers by Chandrasekhar [3], Landau [4], and Oppenheimer and Volkoff [5]. The critical mass $M_{cr}$ is of the order of the solar mass $M_\odot$ for a degenerate Fermi gas of neutrons. It was shown [5] that the equation of state of a relativistic Fermi gas is still far from the ultrarelativistic limit even at the maximum possible mass. It was concluded [6] that above the critical mass, the dominant forces of gravitational attraction make the contraction of matter unbounded. The state of matter in the process of unbounded gravitational contraction (collapse) is referred to as a black hole.

Astrophysical observations point to the existence of supermassive objects at the centers of galaxies. In our Milky Way Galaxy, there is an invisible object located at the same place where the radio source Sagittarius A is. Astronomers estimate its mass from the orbital motion of stars to be as high as $4.3 \times 10^6 \, M_\odot$ and its radius to be less than 0.002 light years [7]. The mass of the Sun is $M_\odot = 1.989 \times 10^{33}$ g. Since the mass of the object at the galactic center exceeds the critical equilibrium mass $M_{cr}$ of a neutron star by six orders of magnitude, black holes are considered as the most likely candidates for supermassive objects at the centers of galaxies.

We have no direct opportunity to observe what is happening beyond the horizon. Nevertheless, it is natural to assume that during unlimited contraction, the pressure rises indefinitely, and the energy per particle will inevitably reach the binding energy of atoms, nuclei, neutrons, and further, step-by-step, "elementary particles". At the same time, the lifetime of galaxies (and, hence, the black holes in their centers) is of the order of the lifetime of the Universe. Since a black hole evolves relatively slowly, local equilibrium concentrations of particles, that enter into "chemical reactions" of transformation into one another, depend on temperature and pressure and are not dependent on specific reaction channels (see [8], §101). One of the most important, but still unclear questions is the inverse influence of the transformation of particles into one another in the process of gravitational self-contraction. The fact that the lifetime of a galaxy (with a collapsing object in the center) is of the order of the lifetime of the Universe suggests that the transformations of particles into one another can slow down the contraction process or even completely stop it. That is why it makes sense to search for and analyze static configurations of gravitating objects in general relativity.

Black holes, neutron stars, and currently, exotic quark [9,10] and boson [11–14] stars are considered as the objects whose structure is determined by the equilibrium state in their own gravitational field. In the process of gravitational collapse, the density of matter increases continuously. At a relatively small pressure, due to additional elasticity in view of the Pauli exclusion principle, Fermi systems look more capable of resisting the gravitational contraction. However, in the process of unbounded contraction, the stage when massive Bose particles (Z bosons, W bosons, Higgs scalar bosons, and/or coupled pairs of fermions) are dominating is inevitable. At low temperatures, a degenerate Bose gas is

energetically more beneficial than a degenerate Fermi gas. At zero temperature, bosons are located at the ground level. This state of bosonic matter is named the Bose–Einstein condensate.

## 1.2. Gravitating Bose–Einstein Condensate

In the frames of the Einstein hypothesis (always det $g_{ik} < 0$), the properties of the equilibrium state of a gravitating Bose–Einstein condensate (in comparison with a degenerate Fermi gas) were summarized in my review article [15]. There is a specific feature of a gravitating Bose–Einstein condensate. The energy spectrum of gravitating bosons is a function of the wave function of the entire condensate. The very presence of a level depends on its population. In particular, at zero temperature for each level, there is a critical total mass $M_{cr}$ above which an equilibrium configuration (and, hence, this level itself) does not exist. For bosons with a rest mass $m \sim 100 \, GeV/c^2$, the critical mass $M_{cr0}$ at the zeroth quantum level is about one million tons. The critical mass $M_{crn}$ increases proportionally to the level number $n$. At $M > M_{crn}$, the next level $n + 1$ acts as the ground state. The concept of the ground state of a boson system is modified. The radius of the sphere occupied by the condensate also increases proportionally to the level number. Therefore, the density does not grow with increasing condensate mass. As long as the spacing between nearby energy levels is large compared to the temperature, no constraints on the total mass arise. However, an equilibrium state on a high energy level $n \gg 1$ is metastable. One bunch of bosons at a high quantum level with a large mass is energetically less favorable than several isolated centers, with a condensate at the zeroth quantum level being in each of them [15].

Under the restriction det $g_{ik} < 0$, the metric component $g^{rr}(r)$, starting from $-1$ at the center, grows with increasing $r$, achieves its maximum value, and decreases back to $-1$ as $r \to \infty$. The function $g^{rr}(r)$ remains negative within the whole interval $0 < r < \infty$. There is no gravitational radius $r_g$ where $g^{rr}(r_g) = 0$. Schwarzschild's $g^{rr}(r) = -1 + \frac{r_g}{r}$ is simply an asymptote applicable at $r \gg r_g$. This property is inherent in equilibrium gravitating clusters of both bosons and fermions. The existence of a finite critical mass $M_{cr}$ is a direct consequence of the restriction det $g_{ik} < 0$.

On the other hand, there are no visible restrictions on the total mass of a point-like gravitating object in the Schwarzschild solution. However, $g^{rr}$ is positive at $0 < r < r_g$, so that the regularity condition in the center $g^{rr}(0) = -1$ is not satisfied. Schwarzschild's solution could be regarded as an approximation of the field produced by a body whose size is small compared to the distance from the observer. To satisfy the requirement of regularity in the center, i.e., $g^{rr}(0) = -1$, there should be another gravitational radius inside the gravitating body, where $g^{rr}$ changes sign also. I am not aware of such a regular static solution, compatible with Einstein's hypothesis (det $g_{ik} < 0$ everywhere).

## 1.3. Regularity Instead of det $g_{ik} < 0$

The determinant det $g_{ik}$ is not an invariant. I allowed myself to analyze what would happen if, **instead of the requirement** det $g_{ik} < 0$, **a weaker condition of regularity is applied**. Allowing det $g_{ik}$ to change sign, I reconsidered the equilibrium structure of a spherically symmetric gravitating Bose–Einstein condensate [16]. By not using the representation $g^{rr} = -e^{-\lambda}$ that fixes the sign, I avoid the trouble of the incompleteness of Schwarzschild's coordinate system [17–19]; see also [20] §103 and [21] §14.

From the point of view of equilibrium in its own gravitational field, it is implied that the number of particles is large, and all interactions, except the gravitational one, are not significant. At the same time, it is assumed that gravitation does not violate the exchange interaction of elementary particles of the Standard Model.

My detailed analysis [16] confirmed the existence of static solutions where $g^{rr}$ changes its sign twice: $g^{rr}(r) = 0$ at $r = r_g$ and at $r = r_h > r_g$; wherein $g_{00}(r_g) \neq 0$. The signature of the metric tensor is changed to $(+, +, -, -)$ within the spherical layer $r_g < r < r_h$. Outside this layer, at $r < r_g$ and $r > r_h$, the signature is $(+, -, -, -)$ as usual.

Nature gave a wonderful gift for us: at absolute zero, the wave function of the ultra quantum condensate of neutral bosons is a classical scalar field (see [22] §30). The wave function, as a scalar field, satisfies the Klein–Gordon equation with $g^{rr}$ being the coefficient at the highest derivative. The Klein–Gordon equation is not determined at $r = r_g, r_h$. Gravitational radii are the boundaries of the intervals where the Klein–Gordon equation satisfies the theorem of existence and uniqueness. The boundary conditions at $r \to r_g - 0$ and $r \to r_g + 0$ can be chosen so that the gravitational radius $r_g$ inside the gravitating object is a regular point. In this case, the second gravitational radius $r_h > r_g$ has to be the termination point. The sphere $r = r_h$ is the border of the volume, occupied by the gravitating matter. For a remote observer, $r_h$ is the event horizon.

If $g^{rr} = 0$ on the surface of a gravitating body, then it follows directly from the Einstein equations that the energy density vanishes at $r = r_h$, and the pressure does not. Definitely, a static state of matter with a high pressure on the interface with a vacuum cannot exist. However, a pressure balance is possible at the interface with the surrounding dark matter. The amount of dark matter in the Universe, according to existing estimates, is several times larger than the amount of ordinary baryonic matter. Using a longitudinal vector field, it is possible to describe adequately the observed manifestations of dark matter, including the galaxy rotation curves [23] and the bending of light by dark matter on the periphery of a galaxy [24]. The balance of pressure on the interface allows connecting the parameters of a black hole with the parameters of dark matter. In particular, the speed on the plateau of a galaxy rotation curve as a function of the black hole's mass is determined; see [16], Formula (68).

Considering only the dominant gravitational interaction, we are not taking into account the non-ideality of the Bose gas, as well as the possible transformation of the bosons into some other particles. As a result, with no account of elasticity, the amplitude of the scalar field diverges logarithmically at $r \to 0$ [16]. This divergence does not affect the mass appearing in the Schwarzschild asymptote, which is visible to a remote observer. At present, we do not know what the bosons of the Standard Model could be turned into with a further increase in pressure. What we can do is to take into account the non-ideality of boson gas using the so-called "model $\lambda \psi^4$". The equilibrium state of a gravitating Bose–Einstein condensate with the account of its elasticity is considered in this article.

## 2. Behind the Horizon

Getting away from Einstein's hypothesis, I do not use the sign-fixing representation $g_{rr} = -e^{\lambda}$ in the spherically symmetric metric:

$$ds^2 = g_{00}(r)\left(dx^0\right)^2 + g_{rr}(r) dr^2 - r^2\left(d\vartheta^2 + \sin^2 \vartheta \, d\varphi^2\right). \tag{1}$$

Generally speaking, the derivation of the Einstein equations inside the interval $r_g < r < r_h$, where $\det g_{ik} > 0$, and outside, where $\det g_{ik} < 0$, should have been carried out separately. In both cases, the static gravitational field, created by a spherically symmetric distribution of matter, satisfies the same Einstein equations (see (100.4) and (100.6) in [20]). We write them down in the form:

$$\left(g^{rr}\right)' + \frac{1 + g^{rr}}{r} = \kappa r T_0^0, \tag{2}$$

$$g^{rr}\left(\frac{1}{r} - \frac{\left(g^{00}\right)'}{g^{00}}\right) + \frac{1}{r} = \kappa r T_r^r. \tag{3}$$

Gravitational constant $\kappa = (8\pi k)/c^4 = 2 \times 10^{-48}$ s²/g × cm, $k = 6.67 \times 10^{-8}$ cm³/g × s². Relation:

$$\frac{\left(g^{rr}\right)'}{g^{rr}} + \frac{\left(g^{00}\right)'}{g^{00}} = \frac{\kappa r}{g^{rr}}\left(T_0^0 - T_r^r\right) \tag{4}$$

follows directly from the Einstein Equations (2) and (3).

The general solution to Equation (2) is:

$$g^{rr}(r) = \frac{\kappa}{r} \int_{r_0}^{r} T_0^0(x) x^2 dx - 1. \tag{5}$$

$r_0$ is a constant of integration. $g^{rr}(r)$ is a regular function, provided that the integral in (5) converges. The convergence of the integral in Equation (5) means that the mass within the layer $(r_0, r)$:

$$M(r_0, r) = \frac{4\pi}{c^2} \int_{r_0}^{r} T_0^0(x) x^2 dx \tag{6}$$

is finite. If the total mass $M = M(0, \infty)$ is finite, the metric component $g^{rr}(r)$ is a smooth continuous function within the whole space $0 < r < \infty$. The regularity of $g^{rr}(r)$ follows from the finiteness of the total mass of matter, regardless of its physical nature. From the convergence of the integral (6) as $r_0 \to 0$, it follows that:

$$g^{rr}(0) = -1. \tag{7}$$

When (7) is fulfilled, the ratio of the circumference to the diameter tends to $\pi$ at $r \to 0$ (a necessary condition for a regular center).

In the Schwarzschild solution [1]:

$$g^{rr}(r) = -1 + \frac{r_h}{r}. \tag{8}$$

At $r \gg r_h$, Newton's law is applicable, and the Schwarzschild radius $r_h$ is proportional to the visible mass $M$ of a gravitating point-like object:

$$M = \frac{c^2}{2k} r_h. \tag{9}$$

However, it does not satisfy the regularity condition (7) in the center. Actually, (8) is applicable outside a spherically symmetric gravitating body, including its boundary, provided that above the surface $T_0^0 = 0$ at $r > r_h$. From Schwarzschild's solution (8), we get the derivative:

$$(g^{rr})' = -\frac{1}{r_h}, \quad r = r_h. \tag{10}$$

Substituting it into Equation (2), we confirm that the energy density vanishes on the surface $r = r_h$:

$$T_0^0(r_h) = 0. \tag{11}$$

On the contrary, as follows from Equation (3), the pressure remains finite on the surface:

$$T_r^r(r_h) = \frac{1}{\kappa r_h^2}. \tag{12}$$

It is not clear how a medium can exist in a static equilibrium state with uncompensated pressure on the interface with a vacuum. Apparently, this is the most likely reason why no static solutions without limitation on mass had yet been found. Nevertheless, an equilibrium state of a black hole becomes possible due to the presence of dark matter. The balance of pressures at the interface of a black hole and dark matter is able to support a static equilibrium of these two phases [16].

## 3. Gravitating Scalar Field

The Lagrangian of a complex scalar field $\psi$ in a curved space-time with the metric tensor $g_{ik}$ (1) has the form:

$$L = g^{ik}\psi^*_{,i}\psi_{,k} - U(\psi^*\psi), \quad U(0) = 0. \tag{13}$$

In accordance with the least action principle, $\psi$ and $\psi^*$ satisfy the Klein–Gordon equation:

$$\frac{1}{\sqrt{|\det g_{ik}|}}\left(\sqrt{|\det g_{ik}|}g^{lm}\psi_{,l}\right)_{;m} = -\frac{\partial U}{\partial |\psi|^2}\psi. \tag{14}$$

Klein–Gordon Equation (14) is invariant against changing the sign of the determinant of the metric tensor, because $\sqrt{-1}$ in the numerator and denominator cancel each other.

Considering the equilibrium state of matter in its own gravitational field, people take into account dominating gravitational forces, and ignore all other interactions. In the application to an ultra quantum Bose–Einstein condensate, I believe that a strong gravitational field does not violate the exchange interaction of bosons.

The main characteristic determining the gravitational properties of a scalar field is the mass of a quantum $m$. In power series of the potential:

$$U\left(|\psi|^2\right) = U(0) + \frac{\partial U}{\partial |\psi|^2}|\psi|^2 + \frac{1}{2}\lambda|\psi|^4 + \dots \tag{15}$$

$\frac{\partial U}{\partial |\psi|^2}$ is a constant having the dimension cm$^{-2}$, and related to the mass of the quantum $m$: $\frac{\partial U}{\partial |\psi|^2} = \left(\frac{mc}{\hbar}\right)^2$.

The terms $\lambda|\psi|^4$ and higher degrees are corrections for collisions of particles and/or other interactions of a non-gravitational nature. Without these terms, the wave function diverges logarithmically as $r \to 0$. The logarithmic divergence of the wave function in the center takes place because, in view of dominating gravity, the interactions of another physical nature are not taken into account.

In the paper [11], within the framework of Einstein's hypothesis, the term $\lambda|\psi|^4/4$ was added to the Lagrangian in order to take into account the non-gravitational pair interaction of bosons. It was shown that the equilibrium configuration in this model differs noticeably from the case of non-interacting bosons even when $\lambda \ll 1$. In the Gross–Pitaevski equation [25,26] for the wave function of the laboratory Bose–Einstein condensate in rubidium vapor [27] and in sodium vapor [28], the corresponding nonlinear term is associated with pair collisions of bosons. As applied to the rarefied gas of atoms of the laboratory Bose–Einstein condensate, $\lambda$ is the coupling constant, expressed in terms of the s-scattering amplitude (length) $a$ of bosons by one another: $\lambda = 4\pi\hbar^2 a/m$ [29]. Since there is no reason to consider the gravitating Bose–Einstein condensate as a rarefied boson gas, $\lambda$ is not reduced to pair collisions only. $\lambda$ is a model parameter characterizing the physical properties of the medium, including its elasticity. With the account of elasticity, the potential (15) is:

$$U\left(|\psi|^2\right) = \left(\frac{mc}{\hbar}\right)^2|\psi|^2 + \frac{1}{2}\lambda|\psi|^4. \tag{16}$$

Time is a cyclic coordinate in a static field. The energy of a single quantum $E = \hbar\omega$ is the integral of motion. For a weak field in a flat space-time, Klein–Gordon Equation (14) is a linear one. Its solution is a plane wave $\psi\left(x^i\right) = \psi_0 \exp\left(i\left(\mathbf{pr} - Et\right)/\hbar\right)$, describing the motion of a particle with the relativistic spectrum $E^2 = \mathbf{p}^2 c^2 + m^2 c^4$. In a curved space-time, $E$ is the conserved energy of the field per one quantum. A scalar field in the state of definite energy $E$ has the form:

$$\psi_E\left(x^i\right) = e^{-iEx^0/\hbar c}\psi(r).$$

Radial wave function $\psi(r)$ obeys the equation:

$$g^{rr}\psi'' + \left( (g^{rr})' + \frac{(\det g_{ik})'}{2\det g_{ik}} g^{rr} \right) \psi' = \left[ \frac{1}{\hbar^2 c^2} \left( g^{00} E^2 - m^2 c^4 \right) - \lambda \left| \psi \right|^2 \right] \psi. \tag{17}$$

Note that $g^{rr}$ is the coefficient at the highest derivative in Equation (17). The Klein–Gordon equation is not defined on hyperspheres $r = r_g, r_h$, where $g^{rr} = 0$.

The Lagrangian (13) of a scalar field does not depend on the derivatives of the metric tensor. The energy-momentum tensor is derived by the formula:

$$T_{ik} = \frac{2}{\sqrt{\left| \det g_{ik} \right|}} \frac{\partial}{\partial g^{ik}} \left( \sqrt{\left| \det g_{ik} \right|} L \right).$$

The components of the mixed energy-momentum tensor, acting in Einstein Equations (2) and (3), are:

$$\begin{aligned}
T_0^0 &= \frac{1}{\hbar^2 c^2} \left( g^{00} E^2 + m^2 c^4 \right) \left| \psi \right|^2 + \frac{1}{2} \lambda \left| \psi \right|^4 - g^{rr} \left| \psi' \right|^2, \\
T_r^r &= \frac{1}{\hbar^2 c^2} \left( -g^{00} E^2 + m^2 c^4 \right) \left| \psi \right|^2 + \frac{1}{2} \lambda \left| \psi \right|^4 + g^{rr} \left| \psi' \right|^2.
\end{aligned} \tag{18}$$

The relation:

$$T_0^0 - T_r^r = \frac{2 g^{00} E^2}{\hbar^2 c^2} \left| \psi \right|^2 - 2 g^{rr} \left| \psi' \right|^2 \tag{19}$$

follows from (18). Enthalpy $\varepsilon + p = T_0^0 - T_r^r$ does not depend on $\lambda$. In the model $\lambda \left| \psi \right|^4$, only elastic collisions of particles are taken into account. Dissipative processes are ignored, and as a result, the gravity and collisions of particles do not change the heat function of the system as a whole.

It is convenient to reduce the set of Klein–Gordon and Einstein Equations (17), (2) and (3) to the normal form, using the relations (4) and (19). In the dimensionless variables:

$$\begin{aligned}
x &= \frac{mc}{\hbar} r, \quad u(x) = \sqrt{\kappa} \psi(r), \quad w(x) = \frac{\hbar \sqrt{\kappa}}{mc} g^{rr}(r) \frac{d\psi}{dr}, \\
g(x) &= g^{rr}(r), \quad h(x) = \frac{E^2}{m^2 c^4} g^{00}(r).
\end{aligned} \tag{20}$$

we have a system of four first-order equations, resolved with respect to derivatives:

$$\frac{du}{dx} = \frac{w}{g}, \tag{21}$$

$$\frac{dw}{dx} = \left( h - 1 - \Lambda u^2 \right) u - \left( \frac{2}{x} - \frac{x}{g} \left( hu^2 - \frac{w^2}{g} \right) \right) w, \tag{22}$$

$$\frac{dg}{dx} = x \left( (h+1) u^2 - \frac{w^2}{g} + \frac{1}{2} \Lambda u^4 \right) - \frac{1+g}{x}, \tag{23}$$

$$-\frac{dv}{dx} = \frac{dh}{h dx} = \frac{1}{x} \left( 1 + \frac{1}{g} \right) - \frac{x}{g} \left( (1-h) u^2 + \frac{w^2}{g} + \frac{1}{2} \Lambda u^4 \right). \tag{24}$$

This set of equations contains one dimensionless parameter:

$$\Lambda = \frac{\hbar^2}{m^2 c^2} \frac{\lambda}{\kappa}$$

characterizing the elasticity of the condensate. In terms of dimensionless functions, the active components of the energy-momentum tensor (18) are:

$$\begin{aligned}
T_0^0 &= \frac{m^2 c^2}{\hbar^2 \kappa} \left[ (h+1) u^2 + \frac{1}{2} \Lambda u^4 - \frac{w^2}{g} \right], \\
T_r^r &= \frac{m^2 c^2}{\hbar^2 \kappa} \left[ (-h+1) u^2 + \frac{1}{2} \Lambda u^4 + \frac{w^2}{g} \right].
\end{aligned} \tag{25}$$

The rest energy of massive bosons of the Standard Model is of the order of 100 GeV. Based on the bosons of the Standard Model, for quantitative estimates, I am going to assume the boson rest mass to be:

$$m = 1.78 \times 10^{-22} g. \tag{26}$$

The corresponding de Broglie wavelength $\lambda$ is:

$$\lambda = \frac{\hbar}{mc} = 1.8 \times 10^{-16} cm. \tag{27}$$

The scales of the energy density $\varepsilon$, and particle density $\rho$ are of the order:

$$\varepsilon \sim \frac{m^2 c^2}{\hbar^2 \kappa} = 1.426 \times 10^{79} \frac{g}{cm \times s^2}, \tag{28}$$

$$\rho \sim \frac{m}{\hbar^2 \kappa} = 8.9 \times 10^{79} \frac{1}{cm^3}. \tag{29}$$

## 4. In the Vicinity of a Gravitational Radius

Denote the dimensionless gravitational radii:

$$x_g = (mc/\hbar)\, r_g \text{ and } x_h = (mc/\hbar)\, r_h. \tag{30}$$

Equations (21)–(24) are defined separately in three intervals: $0 < x < x_g$, $x_g < x < x_h$, and $x_h < x$. They are not defined at $x = x_g$ and at $x = x_h$ because, on these hyperspheres, the coefficient at the highest derivative in the scalar field Equation (17) is zero. Hyperspheres $x = x_g$ and $x = x_h$ are the outer boundaries of intervals where Equations (21)–(24) satisfy the theorem of existence and uniqueness.

It follows from Equation (24) that at $x \to x_g \pm 0$ (+0 means from above and −0 from below):

$$\left(\frac{w^2}{g}\right)_{x \to x_g \pm 0} \to \frac{1}{x_g^2} - u_{g\pm}^2\left(1 - h_{g\pm}\right) - \frac{1}{2}\Lambda u_{g\pm}^4. \tag{31}$$

Here, $u_{g\pm} = \lim\limits_{x \to x_g \pm 0} u(x)$, $h_{g\pm} = \lim\limits_{x \to x_g \pm 0} h(x_g)$ are one-sided limits either from above or from below. The energy-momentum tensor (18) is regular at $r = r_g$ where $g^{rr} = 0$:

$$\begin{aligned} T_0^0\left(r_g\right) &= \frac{1}{\hbar^2 c^2}\left(g^{00}E^2 + m^2 c^4\right)|\psi|^2 + \frac{1}{2}\lambda\,|\psi|^4, \\ T_r^r\left(r_g\right) &= \frac{1}{\hbar^2 c^2}\left(-g^{00}E^2 + m^2 c^4\right)|\psi|^2 + \frac{1}{2}\lambda\,|\psi|^4. \end{aligned} \tag{32}$$

For this reason, the dimensionless wave function $u(x)$ should be continuous at $x = x_g : u_{g+} = u_{g-} \equiv u_g$. Substituting (31) into Equation (23), we obtain:

$$g_g' \equiv g'\left(x_g\right) = 2x_g\left(u_g^2 - \frac{1}{x_g^2} + \frac{1}{2}\Lambda u_g^4\right). \tag{33}$$

Without elasticity ($\Lambda = 0$), the derivative (33) reduces to Equation (29) in [16]. With the account of elasticity, or without, $g(r)$ is a linear regular function at the gravitational radius $r = r_g$. The derivative $g'\left(x_g\right)$ does not depend on the behavior of $w(x)$ and $h(x)$ at $x \to x_g$. The assumption of linearity for $g(x)$ at $x \to x_g$ is not required in advance.

It follows from Equations (31) and (33) that:

$$w^2(x) = \left[\frac{1}{x_g^2} - u_g^2\left(1 - h_{g\pm}\right) - \frac{1}{2}\Lambda u_g^4\right] g'\left(x_g\right)\left(x - x_g\right), \quad x \to x_g \pm 0. \tag{34}$$

On the left-hand side, $w^2 > 0$; thus, the right-hand side of (34) is also a positive quantity. The derivative $g'(x_g)$ in (33) is a non-zero constant. The factor $(x - x_g)$ in (34) changes sign at $x = x_g$. Therefore, the combination $\frac{1}{x_g^2} - u_g^2 (1 - h_{g\pm}) - \frac{1}{2}\Lambda u_g^4$ in square brackets has different signs at $x < x_g$ and $x > x_g$. As far as $g'(x_g) > 0$, the expression (34) makes sense if $h_{g+} \geq 1 - \frac{1}{x_g^2 u_g^2} + \frac{1}{2}\Lambda u_g^2$ at $x > x_g$, and $h_{g-} \leq 1 - \frac{1}{x_g^2 u_g^2} + \frac{1}{2}\Lambda u_g^2$ at $x < x_g$. Therefore, the combination $\frac{1}{x_g^2} - u_g^2 (1 - h_{g\pm}) - \frac{1}{2}\Lambda u_g^4$ either vanishes at $x = x_g$ or its sign is different at $x < x_g$ and at $x > x_g$. In the latter case $h_{g+} \neq h_{g-}$, and $h(x)$ has a gap. As applied to our two gravitational radii (30), $x_g$ is a regular gravitational radius and $x_h > x_g$ is the termination point.

## 4.1. Regular Gravitational Radius

Since the components $T_0^0$ and $T_r^r$ (32) of the energy-momentum tensor have no singularity at $r = r_g$, it is natural to assume that the component $g^{00}$ of the metric tensor is a continuous function:

$$h_{g-} = h_{g+} \equiv h_g = 1 - \frac{1}{x_g^2 u_g^2} + \frac{1}{2}\Lambda u_g^2. \tag{35}$$

Moreover, the functions $u(x)$, $w(x)$ are continuous at $x = x_g$ also. In accordance with (31), $w^2/g = 0$ at $x = x_g$. In addition to $g'_g$ (33), we find the derivatives $u', w'$, and $h'$ at $x = x_g$ from Equations (22) and (24):

$$u'_g = \frac{w'_g}{g'_g}, \quad w'_g = -2u_g \left( \frac{1}{x_g^2 u_g^2} + \frac{1}{2}\Lambda u_g^2 \right), \quad h'_g = -\frac{2h_g}{x_g}\left( \frac{2}{x_g g'_g} - 1 \right). \tag{36}$$

The connection (35) separates a regular solution to Equations (21)–(24), continuous at the interface $x = x_g$ between the regions of different signatures of the metric tensor:

$$u(x) = u_g + u'_g \times (x - x_g), \quad w(x) = w'_g \times (x - x_g),$$
$$g(x) = g'_g \times (x - x_g), \quad h(x) = h_g + h'_g \times (x - x_g), \quad \left| \frac{x}{x_g} - 1 \right| \ll 1. \tag{37}$$

It is convenient to use these relations as boundary conditions in the close vicinity of the interface $x = x_g$. The numerical integration has to be carried out separately for both sides.

There are three free dimensionless parameters $x_g$, $u_g$, and $\Lambda$. Parameter $\Lambda$ characterizes the elasticity of the condensate. In the model "$\lambda\psi^4$", dimensionless $\Lambda$ is a constant, independent of the density. One of the two remaining degrees of parametric freedom has to be used to ensure regularity in the center: $g(0) = -1$. The last degree of freedom allows determining the equilibrium structure of the gravitating Bose–Einstein condensate as a function of its total mass. The search for a regular static solution to the system of Equations (21)–(24) for a gravitating condensate of finite total mass is a nonlinear eigenvalue problem.

## 4.2. Event Horizon

Since the roles of all three degrees of parametric freedom have already been distributed, the second dimensionless gravitational radius $x = x_h > x_g$ has to be the termination point: in accordance with (11), $x_h$ is the dimensionless radius of the sphere bounding the volume occupied by the condensate. For a remote observer, $x_h$ is the event horizon.

The area $x_g < x < x_h$ is a layer with the violated signature $(+, +, -, -)$ of the metric tensor. $g(x) > 0$ in this zone. In this interval, the functions $u(x), w(x), g(x)$, and $h(x)$ satisfy Equations (21)–(24) with boundary conditions (37). At $x = x_h$, the energy density vanishes (11). Comparing $g'(x_h) = -1/x_h$ (10) with (33), valid for both cases $x = x_g$ and $x = x_h$, we find that at the termination point $x = x_h$ (on the event horizon), the scalar field $u_h$ is nonzero:

$$u_h^2 \equiv u^2\left(x_h\right) = \frac{1}{2x_h^2}. \tag{38}$$

Since the radius $r_h$ of the horizon is proportional to the total mass $M$ of the condensate (9), the density of a condensate at the termination point is inversely proportional to the square of total mass, $|\psi|^2 \sim M^{-2}$. On the surface of a supermassive black hole, the term $\frac{1}{2}\lambda|\psi|^4$ in the potential (16) is a negligible correction in view of (38).

According to (12), the pressure on the surface is also inversely proportional to the square of total mass. However, the pressure does not vanish at $r = r_h$. A static state of a Bose–Einstein condensate with an uncompensated sharp surge of pressure at the interface with a vacuum cannot exist. However, this is without dark matter. The presence of dark matter outside the condensate can make it possible to ensure the pressure balance at the interface of these two media [16]. The pressure balance allows establishing a connection between the parameters of a black hole and dark matter. In particular, the dependence of the plateau velocity of a galaxy rotation curve on the mass of a black hole is determined; see (68) in [16].

At $x_h \gg x_g$ in the close vicinity of the event horizon $g\left(x\right) = 1 - \frac{x}{x_h}$, its derivative $g'\left(x_h\right) = -1/x_h$ (10), the metric component $h_h \equiv h\left(x_h\right) \ll 1$, and $|u_h| \ll u_g$. In the close vicinity of the event horizon $1 - \frac{x}{x_h} \ll 1$, Equations (21), (22), and (24) are simplified. The density $u^2\left(x\right)$ terminates with a nonzero value in a square root manner:

$$u^2\left(x\right) = \frac{1}{2x_h^2} - \frac{2}{x_h}\sqrt{1 - \frac{x}{x_h}}, \quad x \to x_h - 0.$$

At $x \to x_h - 0$, the logarithmic derivative of the metric component $g_{00} = e^\nu$ grows as the inverse square root:

$$\nu'\left(x\right) = -\frac{h'\left(x\right)}{h_h} = \frac{2}{\sqrt{1 - \frac{x}{x_h}}}, \quad h_h \ll 1. \tag{39}$$

Gravitational force, acting on a test body, is proportional to $\nu'\left(x\right)$ ([20], Problem 1 at the end of §88). It is directed toward the center. The sphere $x = x_h$ is impenetrable from the inside.

The density and pressure of the condensate decrease with distance from the center. The lifetime of the Standard Model bosons drops down with decreasing pressure. Therefore, the concentration of dominant bosons decreases with the distance from the center. The composition of matter within a black hole is a continuous quark-to-neutron transition from the center toward the surface. Changes in the composition of matter are not taken into account by the "$\lambda\psi^4$" model. If the composition in the near-surface zone differs from the central one, then the sharp root-like growth of $\nu'\left(x\right)$ at $x \to x_h - 0$ (39) can become smoothed out. Some more features of the composition of a black hole depending on its mass are discussed below in Section 5. The structure of the shell of a black hole is a worthy topic for further consideration.

## 5. Regular Static Solutions with no Restriction on Mass

### 5.1. Upper Boundary of the Regularity Strip: Simple Analytic Solution

Numerical analysis confirms that with the account of elasticity, a necessary condition for the existence of solutions with finite mass is $\frac{h'_g}{h_g} < 0$. By virtue of the relations (33) and (36), this condition restricts $x_g^2$ both from below and from above:

$$\frac{1}{2} < \left(\frac{x_g}{x_{g\,\text{max}}}\right)^2 < 1, \quad x_{g\,\text{max}}^2\left(u_g, \Lambda\right) = 2\left(u_g^2 + \frac{1}{2}\Lambda u_g^4\right)^{-1}. \tag{40}$$

Without elasticity ($\Lambda = 0$), the inequalities (40) reduce to the strip $1 < x_g^2 u_g^2 < 2$ in the plane of parameters $x_g, u_g^2$; see Figure 1 in [16].

Klein–Gordon Equation (14) is a nonlinear homogeneous equation. The trivial solution to the set (21)–(24) $u = 0$, $w = 0$, $g = -1$, $h = h_0$ corresponds to a flat space-time in the absence of a scalar field. The constant $h_0$ is the freedom for the choice of time units.

There is a nontrivial regular analytic solution to the system of dimensionless Klein–Gordon and Einstein Equations (21)–(24):

$$u(x) = u_0, \quad w(x) = 0, \quad g(x) = -1 + u_0^2 x^2/3, \quad h(x) = 1/3, \quad \Lambda u_0^2 = -2/3. \tag{41}$$

It could be interpreted as if the Universe was uniformly filled by a Bose–Einstein condensate. The total mass $M$ is infinite. Though this solution is unrealizable, it has a physical meaning, and it facilitates finding regular solutions with finite total mass.

The relation:

$$\Lambda u_0^2 = -2/3 \tag{42}$$

fixes the balance of the density $u_0^2$ and elasticity $\Lambda$ of the condensate. The less is the elasticity, the denser the condensate is compressed by its own gravitational field. Within the model "$\lambda\psi^4$", the finite density in the center is determined by the elasticity of the condensate: $u_0^2 = -\frac{2}{3\Lambda}$. Solution (41) is regular in the center: $g(0) = -1$. It corresponds to the upper boundary of the interval (40) of regular solutions with finite mass. Really, according to (41), $g(x) = 0$ at $x^2 = x_g^2 = 3/u_0^2$. With (42) from (40), we get $x_{g\,\max}^2 = 2\left(u_g^2 + \frac{1}{2}\Lambda u_g^4\right)^{-1} = 3u_g^{-2}$. Thus,

$$\left(\frac{x_g}{x_{g\,\max}}\right)^2 = 1. \tag{43}$$

Moreover, at $w = 0$, $h = 1/3$, and $\Lambda u_0^2 = -2/3$ the energy-momentum tensor (25):

$$T_0^0 = \frac{m^2 c^2}{\hbar^2 \kappa} u_0^2, \quad T_r^r = \frac{m^2 c^2}{3\hbar^2 \kappa} u_0^2 \tag{44}$$

corresponds to the ultrarelativistic equation of state $p = \varepsilon/3$.

For further comparisons, the nontrivial regular solution (41) for $u_0^2 = 0.1$ is shown in Figure 1.

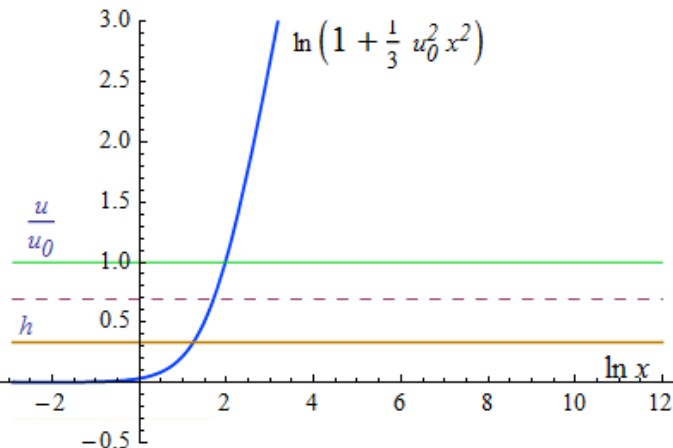

**Figure 1.** A nontrivial solution (41) to Equations (21)–(24) for $u_0^2 = u_g^2 = 0.1$, $\left(\frac{x_g}{x_{g\,\max}}\right)^2 = 1$. A logarithmic scale along the $x$ axis is used. The dashed horizontal line is the level $\ln 2$, where $g(x) = -1 + \frac{1}{3}u_0^2 x^2 = 0$.

In view of (42), the gravitational radius of the nontrivial solution (41) is:

$$x_g = x_{g\,\text{max}} = \sqrt{30}, \ \ \ln x_g = 1,70059869..., \ \ \ u_0^2 = 0.1. \tag{45}$$

I intentionally chose for demonstration a small dimensionless density $u_0^2 = 0.1$, so that all basic properties of a regular static solution are visible on a single graph in Figure 2. Using logarithmic scales along both axes allows me to overlap a huge interval from the de Broglie wave length of a 100 GeV boson (27) to the size of a black hole. To see the whole $g(x) \geq -1$ on a logarithmic scale, I show $\ln(g(x) + 2)$ instead of $\ln g(x)$.

## 5.2. An Illustrative Example of a Regular Static Solution

The slightest deviation from the upper boundary (43) into the interval (40), for instance:

$$\left(\frac{x_g}{x_{g\,\text{max}}}\right)^2 = 0.99995, \tag{46}$$

gives us a regular static solution with finite total mass. Other parameters in Figure 2 are the same as in Figure 1: $u_g^2 = 0.1$, $\Lambda u_g^2 = -2/3$. It is convenient to show the increase of $g(x)$ from $-1$ to the maximum, followed by a decrease back to $-1$, using a double logarithmic scale. In order to see the entire dependence $g(x)$ on a single graph, including the regions where $g(x) < 0$, I moved $g(x)$ two steps up in Figure 2. Like in Figure 1, the dashed horizontal line $\ln 2$ is the level where $g(x) = 0$.

The two dashed vertical lines indicate gravitational radii $x_g = 5.4770886427...$ ($\ln x_g = 1.70057369$), and $x_h = 388.1645$ ($\ln x_h = 5.961$).

A seemingly insignificant deviation $-0.00005$ of $x_g^2$ from the upper boundary (45) causes the growing $g(x)$ to decrease, so that a second gravitational radius $x_h$ appears. For a remote observer, $x_h$ is the event horizon. The condensate wave function $u(x)/u_g$ (green curve in Figure 2) deviates from its constant value in Figure 1, experiences damped oscillations, and terminates at $x = x_h$ with a nonzero value (38). The dimensionless component of the metric tensor $h(x)$ (brown curve in Figure 2) decreases monotonously. $h(x)$ also ends with a non-zero value at $x = x_h$.

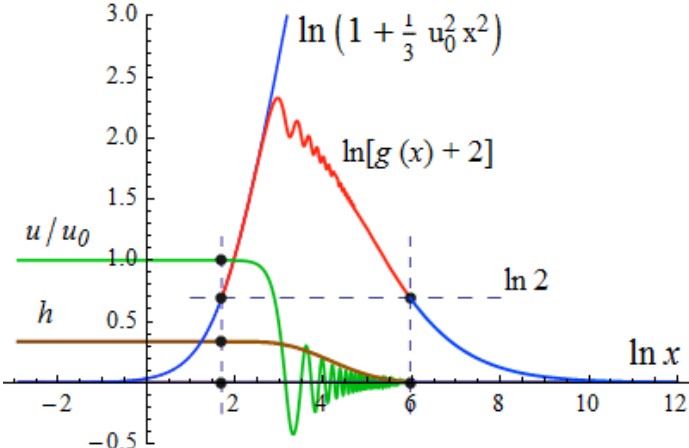

**Figure 2.** The blue growing curve $\ln\left(1 + \frac{1}{3}u_0^2 x^2\right)$ is the same as in Figure 1: $u_g^2 = 0.1$, $\Lambda u_g^2 = -2/3$, $\left(\frac{x_g}{x_{g\,\text{max}}}\right)^2 = 1$. The red curve with the parameters $u_g^2 = 0.1$, $\Lambda u_g^2 = -2/3$, $\left(\frac{x_g}{x_{g\,\text{max}}}\right)^2 = 0.99995$ practically coincides with the blue one on the growing left side. However, even with a so small decrease of $x_g$, the growing part of $g(x)$ changes to a decreasing one, and a second gravitational radius $x_h$ appears.

### 5.3. Total Mass as a Function of Density in the Center

In the model "$\lambda \psi^4$", the balance of density and elasticity, together with the requirement of a regular center $g(0) = -1$, unambiguously connect the parameter $\Lambda$ with the density $u_g^2$ of the condensate (42). the minus sign means that a static equilibrium state is possible only if the non-gravitational interaction of bosons is repulsive. The weaker the repulsion, the stronger the condensate is compressed by its own gravitational field. In the vicinity of the upper border (46) $x_g^2 = 2.99985u_g^{-2}$, the dependence of the event horizon $x_h$ on the density $u_g^2$ is presented in Figure 3. According to Relations (9) and (20):

$$M = \frac{M_{Pl}^2}{2m} x_h.$$

The Plank mass $M_{Pl} = \sqrt{c\hbar/k} = 2.177 \times 10^{-5}$ g. For bosons with the rest mass $m = 1.78 \times 10^{-22}$ g (26), total mass $M$ is:

$$M = 1.33 \times 10^{12} x_h \text{ gram.}$$

The solid line in Figure 3 is the interpolation of red points $x_h\left(u_g^2\right)$ found numerically. Dashed horizontal lines in Figure 3 correspond to the masses of the Earth, Sun, and black hole in the center of our Milky Way, respectively.

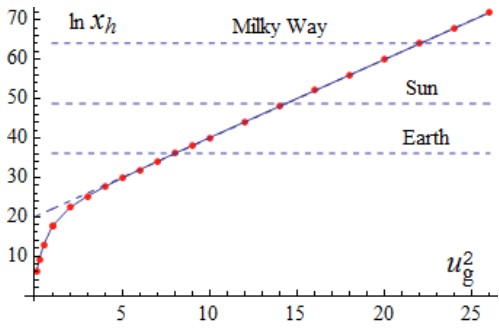

| $u_g^2$ | 0.1 | 0.25 | 0.5 | 1 | 2 | 3 | 4 |
|---|---|---|---|---|---|---|---|
| ln$x_h$ | 5.96 | 9.09 | 12.8 | 17.6 | 22.3 | 25.2 | 27.6 |
| $u_g^2$ | 5 | 6 | 7 | 8 | 9 | 10 | 12 |
| ln$x_h$ | 29.8 | 31.98 | 34.05 | 36.1 | 38.1 | 40.1 | 44.1 |
| $u_g^2$ | 14 | 16 | 18 | 20 | 22 | 24 | 26 |
| ln$x_h$ | 48.1 | 52.1 | 56.07 | 60.0 | 63.96 | 67.92 | 71.88 |

Coordinates of red points

**Figure 3.** Dependence of the event horizon radius $x_h$ on the condensate density $u_g^2$ in dimensionless units. The logarithmic scale is used along the vertical axis. Three dashed horizontal lines correspond to the masses of the Earth, the Sun, and the black hole in the center of the Milky Way, respectfully.

At $u_g^2 \gg 1$, the numerically found points lie on the straight line:

$$\ln x_h = 20 + 2u_g^2, \quad x_g^2 = 2.99985u_g^{-2}.$$

The lowest red point $u_g^2 = 0.1$ in Figure 3 corresponds to the solution shown in Figure 2. Its mass $M \sim 3 \times 10^{-20} M_\odot$ is significantly less than the critical $M_{cr} \sim M_\odot$. Using a black hole of a very small mass as an example, I can clearly demonstrate the basic properties of a regular static state of the gravitating Bose–Einstein condensate.

By the way, it is appropriate to recall that the possibility of the gravitational collapse of a strongly compressed neutron bunch of a small mass has been pointed out by Zel'dovich. His paper [30] is now considered as the first step to understanding the nature of the primary small mass black holes in the early Universe [31].

A static equilibrium state of a gravitating body with a small mass $M < M_{cr} \sim M_\odot$ is ensured by the elasticity of a Fermi gas. To realize the small mass collapse noted by Zel'dovich [30], it is necessary to overcome the potential barrier associated with the elasticity of neutron Fermi gas. This higher is the barrier, the smaller is the total mass $M$ of neutrons in comparison with the critical mass $M_{cr}$.

At $M > M_{cr}$, the elasticity of the Fermi gas of neutrons is insufficient to withstand gravitational compression forces. The potential barrier, associated with the elasticity of Fermi gas, disappears. Objects with mass $M > M_{cr}$ are subject to gravitational collapse. If there were no transformations

of particles into one another with increasing density, the compression would have been unlimited. Due to the mutual transformations of particles, the matter in the state of a Bose condensate remains able to resist compression. The collapse has to be stopped for the same quantum reasons that electrons do not fall down on nuclei in atoms. If a static state of gravitating fermions is ensured by the Pauli exclusion principle, then for bosons, the main role belongs to the Heisenberg uncertainty principle. A regular static solution to the system of Klein–Gordon (17) and Einstein Equations (2) and (3) describes a state where the collapse could be stopped. The dominance of bosonic matter can occur by direct transformations of some particles into others, as well as due to the Cooper effect [32], i.e., pairing of fermions into bosonic quasiparticles. Mutual particle transformations are the result of the interactions of a non-gravitational nature. Being in the ground state at zero temperature, a Bose–Einstein condensate is stable.

### 5.4. Black Holes and Dark Matter

On the border of a black hole $r = r_h$, the pressure (12) is inversely proportional to the mass squared:

$$\left| p\left(r_h\right) \right| = T_r^r\left(r_h\right) = \frac{1}{\kappa r_h^2} = \frac{c^8}{32\pi k^3 M^2}. \tag{47}$$

For a collapsed black hole with the mass $M \sim M_{cr} \sim M_\odot$, the pressure in the center $p\left(r_g\right) \sim 6 \times 10^{73}$ bar, and on the surface (47), $p\left(r_h\right) \sim 5 \times 10^{30}$ bar. At a pressure $\sim 10^{30}$ bar, a neutron Fermi gas would have been ultrarelativistic; see [8], §106. It is natural to assume that at a so high a pressure on the surface, the dominance of the Standard Model bosons extends all the way to the event horizon $r_h$.

With no forces from outside, a static equilibrium of a black hole with a pressure of $10^{30}$ bar on the interface with a vacuum is impossible. In reality, a black hole interfaces with dark matter, and not with a vacuum. The pressure balance at the interface of a black hole with dark matter [16] allowed establishing the connection between the parameters of these two objects. In particular, the dependence of the plateau velocity $V_{pl}$ of a galaxy rotation curve on the mass $M$ of a black hole was established [16]:

$$V_{pl} = c\, \frac{M_{Pl}^2}{4\,\sqrt{\mu m M}}. \tag{48}$$

Here, $M_{Pl}$ is the Plank mass, $\mu$ is the mass of a quantum of the longitudinal vector field describing dark matter, and $m$ is the mass of a quantum of the scalar field, which is the wave function of the Bose–Einstein condensate.

The plateau velocities of rotation curves of the galaxies UMa: NGC 3726 and UMa: NGC 3769, mentioned in the review [33], are $V_{pl\,3726} \approx 150$ km/s and $V_{pl\,3769} \approx 120$ km/s. The period of damping oscillations is about $\lambdabar \approx 13$ kpc for both galaxies; see Figure 5 in [33]. This corresponds to the de Broglie wavelength of a particle with a mass $\mu = \hbar/c\lambdabar \approx 0.7 \times 10^{-60}$ g. The rest mass of a boson is $m \approx 1.78 \times 10^{-22}$ g (26). It follows from (48) that the masses of black holes, located in the centers of galaxies NGC 3726 and NGC 3769 of the Ursa Major cluster, are:

$$M_{3726} \approx 5.2 \times 10^{33}\text{g},\, M_{3769} \approx 6.5 \times 10^{33}\text{g}. \tag{49}$$

The accuracy of the absolute values of black hole masses is rather small because it is not clear what kind of bosons the condensate consists. However, if the internal stuff of the two black holes is the same, then the ratio of masses can be determined more accurately: the errors would depend only on the accuracy of observable plateau velocities.

With the account of dark matter, the velocity $V\left(r\right)$ of a rotating star as a function of the distance $r$ from the center of the galaxy,

$$V(r) = \sqrt{V_{pl}^2 \left(1 - \frac{\sin(2\mu cr/\hbar)}{2\mu cr/\hbar}\right) + \frac{c^2}{2}\frac{r_h}{r}}, \tag{50}$$

was derived in my paper [24]. The plateau velocity $V_{pl}$ in (50) was a free parameter. Now, knowing the dependence of $V_{pl}$ on the mass of the black hole $M$ (48), we can compare the contributions of dark matter $V_{pl}^2\left(1 - \frac{\sin(2\mu cr/\hbar)}{2\mu cr/\hbar}\right) \sim c^2\frac{M_{Pl}^4}{16\mu mM^2}$ and the black hole $\frac{c^2}{2}\frac{r_h}{r} \sim c^2\frac{\mu M}{M_{Pl}^2}$ at $\frac{\mu cr}{\hbar} \sim 1$. The contributions are of the same order if the mass of a black hole $M$ is:

$$M = \widetilde{M} \equiv \frac{M_{Pl}^2}{(16m\mu^2)^{1/3}}. \tag{51}$$

Substituting $m \approx 1.78 \times 10^{-22}g$ and $\mu = 0.7 \times 10^{-60}g$ into (51), we get

$$\widetilde{M} = 4.24 \times 10^{37}g. \tag{52}$$

In galaxies like NGC 3726 and NGC 3769 of the Ursa Major cluster, whose black hole masses (49) are small compared to (52), the rotation of stars is driven by dark matter. In our Milky Way, on the contrary, the mass of the black hole $8.6 \times 10^{39}$ g is two orders of magnitude greater than (52). The motion of stars is regulated by the black hole in the center. In our galaxy, dark matter does not play a noticeable role.

The analysis of the general properties of 240 different galaxies [34,35] shows that the difference between the observed centripetal acceleration and the Newtonian one decreases with increasing acceleration; see Figure 4. The straight 1:1 line shows where the observed and Newtonian accelerations coincide. The larger the mass of a black hole in the center of a galaxy, the bigger the acceleration is.

The decreasing difference in accelerations in Figure 4 is in accordance with (48). Together with the plateau velocity $V_{pl}$, the influence of dark matter decreases with growing mass $M$ of a black hole. The galaxies NGC 3726 and NGC 3769 of the Ursa Major cluster are on the left side in Figure 4, while our Milky Way galaxy is on the right side.

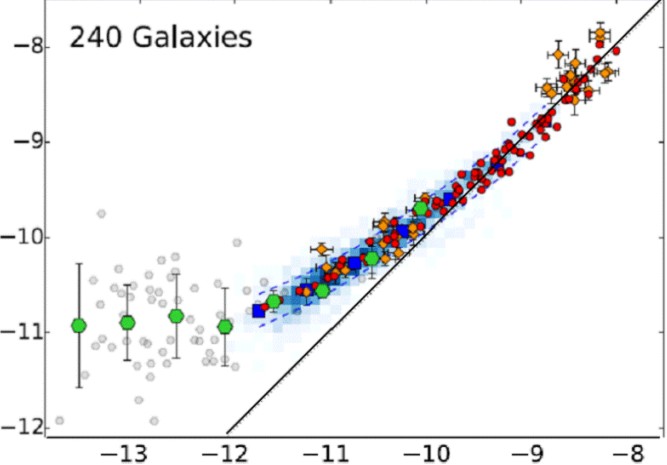

**Figure 4.** The observed acceleration (vertical axis) as a function of the expected one from the distribution of baryons (horizontal axis) for 240 galaxies [34,35]. Logarithmic scales along both axes. Colored dots correspond to galaxies of different morphological types. The straight 1:1 line shows where the observed and the expected Newtonian accelerations would coincide without dark matter.

### 5.5. Superheavy Black Hole

The pressure on the surface of a black hole decreases with increasing mass as $M^{-2}$ (47). The larger the mass of a black hole, the less the pressure on its surface. The mass of the black hole in the center of Milky Way is six orders of magnitude larger than the Sun mass. The pressure in the center is

$p\left(r_g\right) \approx 10^{74}$ bar and, on the surface, $p\left(r_h\right) \approx 3 \times 10^{17}$ bar. At $3 \times 10^{17}$ bar, the Fermi energy of a degenerate electron gas is of the order of $m_e c^2$, and $m_e$ is the electron rest mass. Consequently, the region of dominance of bosonic matter does not reach the surface of this black hole.

From the pressure $10^{74}$ bar in the center to $3 \times 10^{17}$ bar on the surface, the matter of the Milky Way black hole is a continuous quark-to-neutron transition. At present, we do not have an equation of state of matter in the transition between the nuclear regime at "low" pressures and the quark regime at high pressures derived in quantum chromodynamics from first principles. An interpolation model named the "unified equations of state" was considered in the review [10]. The shell structure of a superheavy black hole has to be considered with the account of nuclear and electromagnetic forces. Actually, a static state of a black hole with $M \gg \widetilde{M}$ does not need a support from dark matter. A different composition of matter near the surface of a black hole can smooth out the sharp root behavior of $v'(x)$ (39). The question of the shell structure of a black hole at $M \gg M_{cr}$ is beyond the scope of this paper.

The upper dashed horizontal line in Figure 3 corresponds to the total mass $M \approx 4 \times 10^6 M_\odot$ [7] of the black hole located in the center of our Milky Way. A regular static solution to the system of Klein–Gordon and Einstein Equations (21)–(24), corresponding to this mass, is presented in Figure 5. $u_g^2 = 22$, and $x_g^2 = 2.99985 u_g^{-2} = 0.1363568...$ ($x_g = 0.369..$, $\ln x_g = -0.99624$) are dimensionless parameters of this solution.

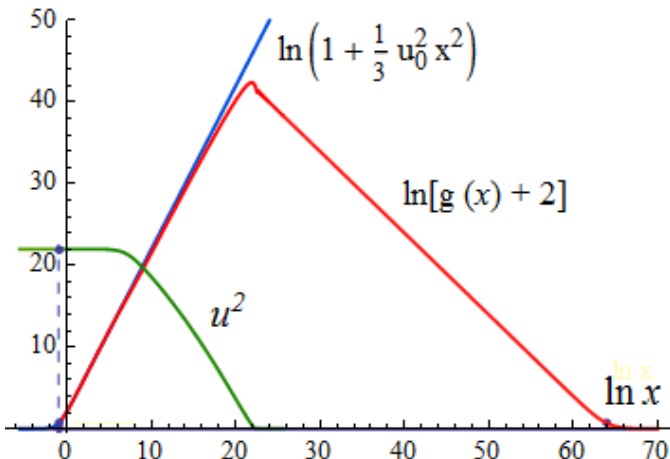

**Figure 5.** Solution to Equations (21)–(24) corresponding to the superheavy black hole located in the center of the Milky Way Galaxy. $u_g^2 = 22$, and $x_g^2 = 2.99985 u_g^{-2}$. The red line is $\ln\left(g\left(x\right) + 2\right)$, and the green line is $u^2\left(x\right)$. The blue line shows the analytical solution $g\left(x\right) = -1 + u_0^2 x^2/3$ (41) on the upper border $x_g^2 = 3 u_g^{-2}$, $u_g^2 = 22$ of the regularity string (40).

$\Lambda = -\frac{2}{3} u_g^{-2} = -\frac{1}{33}$ is the dimensionless elasticity of the condensate. The red line is $g\left(x\right)$ raised two steps up. On the double logarithmic scale, both the growing part $\sim x^2$ and the decreasing part $\sim 1/x$ are straight lines. The growing blue line is $g\left(x\right)$ (41) of the analytical solution at the upper boundary (43). The green line presents the dimensionless density of the condensate $u^2\left(x\right)$. Small details (damped oscillations of $u\left(x\right)$; oscillations of $g\left(x\right)$ at the transition from growth to decrease; and two points on the level $\ln 2 = 0.693$ at $\ln x_g = -0.996$ and $\ln x_h = 63.96$ where $g\left(x_g\right) = g\left(x_h\right) = 0$) are not visible in the huge range from $10^{-16}$ cm to $10^{11}$ cm in Figure 5. This is why I presented Figure 2 for clarity. The topology of a regular static solution to Equations (21)–(24) is the same in both limiting cases: with a huge mass $M \approx 4 \times 10^6 M_\odot$ and with an extremely small mass $M \sim 3 \times 10^{-20} M_\odot$.

Functions $u\left(x\right)$ and $h\left(x\right)$ are presented separately in Figure 6. Parameters $u_g^2 = 22$ and $x_g^2 = 0.99995 x_{g\,\max}^2$ are the same as in Figure 5. $h\left(x\right)$ is a monotonically decreasing function. Function $u\left(x\right)$ decreases with oscillations. Both functions terminate at $x = x_h$ with very small, but still nonzero values. The termination point $\ln x_h = 63.96$ is far outside the graph in Figure 6.

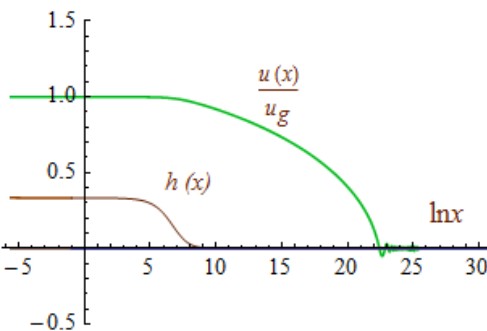

**Figure 6.** Solution to Equations (21)–(24) corresponding to the superheavy black hole located in the center of the Milky Way Galaxy. $u_g^2 = 22$, and $x_g^2 = 2.99985u_g^{-2}$. The green line is $u(x)/u_g$, and the brown line is $h(x)$. The horizon $x_h$ $(\ln x_h = 63.96)$ is far outside this graph.

The metric component $g(x)$ has a maximum at $x = x_m \approx \left(x_g^2 x_h\right)^{1/3}$. For the case presented in Figures 5 and 6, $x_g \approx 0.369$ is of the order of unity. At $x \gg x_m \sim x_h^{1/3}$, the contribution from $u(x)$ and $h(x)$ to the total mass of the condensate is negligible. Therefore, at $x \gg x_h^{1/3}$, metric function $g(x) = \frac{x_h}{x} - 1$. Equations (21) and (22) defining $u(x)$ and $w(x)$ are simplified:

$$\frac{du}{dx} = \frac{x}{x_h}w, \quad \frac{dw}{dx} = -u - \frac{2}{x}w, \quad x_h^{1/3} \ll x \ll x_h. \tag{53}$$

By substituting $x = y^{2/3}$, Equation (53) is reduced to a Bessel equation:

$$\frac{d^2u}{dy^2} + \frac{1}{y}\frac{du}{dy} + \frac{4}{9x_h}u = 0.$$

The general solution to Equation (53) is:

$$u(x) = C_1 \operatorname{BesselJ}_0\left(\frac{2}{3\sqrt{x_h}}x^{3/2}\right) + C_2 \operatorname{BesselY}_0\left(\frac{2}{3\sqrt{x_h}}x^{3/2}\right).$$

In the considered range $x_h^{1/3} \ll x \ll x_h$, Bessel functions are reduced to sines and cosines:

$$u(x) = 0.024\frac{x_h^{1/4}}{x^{3/4}}\left[\sin\left(\frac{2x^{3/2}}{3x_h^{1/2}} + \frac{\pi}{6}\right) - \cos\left(\frac{2x^{3/2}}{3x_h^{1/2}} + \frac{\pi}{6}\right)\right], \quad x_h^{1/3} \ll x \ll x_h. \tag{54}$$

With this appropriate choice of the amplitude and the phase, the expression (54) coincides with $u(x)$ found numerically; see Figure 7.

In the intermediate interval $x_h^{1/3} \ll x \ll x_h$, the condensate density $u^2(x)$ decreases with the distance from the center as $x^{-3/2}$. In the model "$\lambda\psi^4$", which does not take into account the mutual transformations of particles, $u^2$ would terminate at $x = x_h$ with the nonzero value (38). In reality, under pressure $\sim 10^{17}$ bar, the matter is most likely in the state of a relativistic plasma.

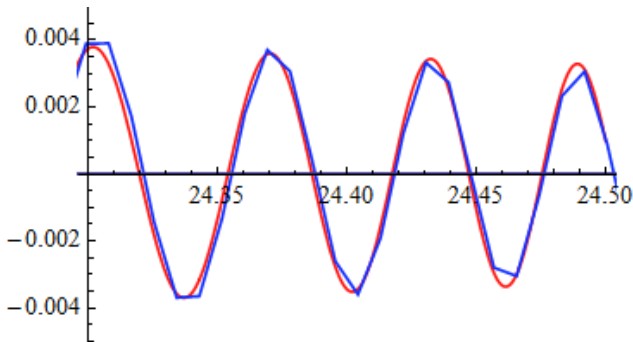

**Figure 7.** Fragment of oscillating $u(x)$ in the interval $24.3 < \ln x < 24.5$ derived analytically (54) (red line) and found numerically (blue line). The angularity of the blue line is due to the fact that the condensing oscillations in the huge interval $(x_m, x_h)$ are obtained at the limit of computer accuracy.

## 6. Conclusions

After I replaced the constraint $\det g_{ik} < 0$ by a weaker requirement of regularity (all invariants of $g_{ik}$ are finite), it became possible to find a regular static spherically symmetric solution to the Einstein equations with no restriction on mass. Regular static solutions to the Einstein equations, with no limitation on mass, describe the state of a black hole to which the gravitational collapse leads. By not using the representation $g^{rr} = -e^{-\lambda}$ that fixes the sign, I avoided long-standing problems, such as: a singularity in the center and the incompleteness of the reference frame in the Schwarzschild solution; how the process of unlimited compression of a black hole can continue throughout the entire lifetime of the Universe. Previously unknown static solutions, describing the states of ultimately compressed black holes, confirm that a static state of matter with the ultra relativistic equation of state does not contradict Einstein's general theory of relativity.

The price for the static solution is the existence of a spherical layer with a broken metric signature $(+, +, -, -)$. This layer is located beyond the event horizon. Though it does not lead to logical contradictions, the physical meaning of a layer with the metric signature $(+, +, -, -)$ still has to be understood.

**Funding:** This research received no external funding.

**Conflicts of Interest:** The authors declare no conflict of interest.

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
