# Peer review of "Guessing the Riddle of a Black Hole"

_universe, doi:10.3390/universe6080113_

Round 1
Reviewer 1 Report
Under finding of regular static solutions to the Einstein equations for a spherically symmetric distribution of matter author uses a "weaker condition of regularity": all invariants of g_{ik} are finite. It makes possible to find regular static solutions with no restriction on total mass.
However for a static solution to exist, in this case a spherical layer is required with a broken metric signature (+,+,-,-). This in my opinion greatly reduces the importance of this work for the physics of black holes. Particles falling into this black hole will be inside such a layer with two time-like and two spatial dimensions. The physics of processes in spaces with two times is unknown. So the author's approach does not give us ``guessing the riddle of a black hole'', what the title of the article promises us.
However, from the point of view of model mathematical research, a number of new results and new solutions have been obtained in the work, which are of some mathematical interest.
In my opinion, in the abstract of the article should be explicitly indicated that the existence of the static solution requires the existence of a layer with a broken metric signature (+,+,-,-). The statement "Long-standing problems, such as singularity in the center and incompleteness of the reference frame in the Schwarzschild solution, and how the process of unlimited compression of a black hole can continue throughout the entire lifetime of the Universe, disappear by themselves" should be deleted from the abstract. There is no physical basis for such a statement.
After making these corrections, the article can be published.
Author Response
Thanks for your reviewing. ENclosed please find the cover letter.

Reviewer 2 Report
Guessing the riddle of a black hole
Report
In this work entitled “Guessing the riddle of a black hole”, the author presents a study on the structure of a extremely compressed Bose-Einstein condensate due to its own gravitational field, and the associated solutions to Einstein equations for the corresponding spherically symmetric distribution which describes the state the of a black hole and their physical properties, including the possible interaction with surrounding dark matter.
Regarding this manuscript, in my opinion, it is scientifically sound, mathematically consistent, and the presented topic is relevant and interesting. I found that in general, it is well written. However, I have found some typos that should be corrected. I highly recommend to the author to perform an extensive revision of the manuscript to attend this issue, preferably by a native English speaker.
Beside these details, in its current form this manuscript presents some problems that must be attended and corrected before it could be adequate to be publish in Universe. In the following I will proceed to present each one of the points that in my opinion, must be addressed before this manuscript could further proceed to be considered for publication.
- My biggest concern with this paper is the following: In the first three sections of this work the author presents several paragraphs that are taken verbatim from one of his own previous works, in particular, the author considers the paper “Static State of a black hole supported by dark matter”. Published in Universe 5, 198 (2019). I understand that the present work is a follow up of this paper, considering this time the quartic order term in the in the power series for the potential U(|\psi|^2) given in eq. (15) of this work, corresponding to elastic collisions between particles; however the author should be very careful in how the information is presented, and to avoid take parts word by word, since it could be even considered self-plagiarism. For instance, the last paragraph of page nine “From the point of view of equilibrium in its own gravitational field, it is implied that the number of quanta of the field is large, and all interactions, except the gravitational one, are not significant. Today, the interaction of dark and ordinary matter is observed through the curvature of space-time only.” is taken verbatim from page 5 of Universe 5, 198 (2019). Therefore, I would like to recommend to the author to heavily edit these sections to avoid this by means of citing, rewriting, etc.
- Introduction section is very long and mainly have an anecdotical nature, even when information presented is relevant in order to justify the use of the condition that all invariants of the metric tensor are finite, its extension should be reduced in order to present the information in a more consisting way.
- Introduction section would be greatly improved if an outline of this work is included on it, this work spans over several topics, therefore it would be important for the reader to have a guide to follow.
- The analysis presented by the author is a purely geometric one, in my opinion, this work would be improved if a thermodynamic analysis on the behavior of such black holes is included or at least hinted.
- On the contrary to the Introduction section, the Conclusions sections is very brief and basically ignores every result presented in this work, I would like to recommend to the author to highlight these results in this section.
These are my main concerns regarding this manuscript, and they must be addressed before it is suitable for publication. Solving properly these problems the overall quality of the article will be greatly improved, and it will be suitable for publication in Universe.
Author Response
Thanks for your reviewing. Enclosed please find the cover letter.

Reviewer 3 Report
The author of the paper universe-874607 has studied static structure of matter, extremely
compressed to the state of a Bose-Einstein condensate
by its own gravitational Â…field. Instead of the widely spread restriction $det g_{ik} < 0$,
a weaker condition of regularity is used. It makes possible to find
regular static solutions to the Einstein field equations for a spherically symmetric distribution of matter
with no restriction on total mass. Though the gravitation dominates at extremely high density, it is assumed that
it does not violate the exchange interaction of elementary particles of the Standard
Model. The obtained regular static solution to Einstein equations, having no limitation on mass, pretends
to describe the end state of a black hole as result of the gravitational collapse. The features of
a collapsed black hole, its internal composition depending on total mass, and the relation with
surrounding dark matter, are considered.
After replacement of the constraint for the determinant of the metric tensor by a weaker requirement of regularity
the author has found regular static spherically-symmetric
solution to Einstein equations with no restriction on mass. The price for the static solution
is the existence of a spherical layer with a broken metric signature which
is located beyond the event horizon.
The calculations performed are standard ones and correct, however, I have the following comments and suggestions:
i) The abstract of the paper has not been well written. The current form of the abstract looks like an introduction to the topic.
It would be better if the author could slightly revise the text of abstract in a form presenting the main results from the
main part of the paper.
ii) The conclusion is very short. I advise the author to summarize in detail the obtained results with disccussion.
iii) The existence of a spherical layer with a broken metric signature and located beyond the event horizon
has to be justified from the physical point of view.
iv) The quality of plots and figures does not meet the standards of the Universe journal and has to be improved.
v) More detailed analysis of the energy conditions and curvature invariants has to be performed towards
detailed description of the obtained singularity free solution.
vi) Since the Universe journal is oriented on publication of the papers related to physics the author has to explore
possible astrophysical applications of the obtained regular solution.
I do not think that the performed mathematical calculations
for the non-singular black hole spacetime would warrant a
publication in a high impact journal like the \textbf{UNIVERSE} specialized in publishing results
having physical and astrophysical importance in the revised form. At the mment
it really contains the mathematical results rather than
physical ones since the spacetime metric has undefined properties and no astrophysical application.
Physics behind these properties is not well defined and in the literature there are many regular solutions.
What is the advantage of using this one?
Since the study of this manuscript is mathematically correct and
indeed of interest in addressing the behaviour of the nonsingular black holes I would recommend to the author
to revise it along my comments and resubmit the revised version to journal. Then I would recommend it for publication in the \textbf{UNIVERSE} journal.
I hope the revised paper would be suitable for publication in the \textbf{UNIVERSE} journal.
\end{document}
Author Response

(The authors gave the same response as above.)

Reviewer 4 Report
The paper "Guessing the riddle of a black hole" deals with some old unsolved puzzles concerning black hole physics. Namely, its internal structure, boundaries and composition. After relaxing the usual requirement of a fixed sign for the metric determinant and using the Klein-Gordon equation to describe the matter source dynamics, the author is able to find a regular solution composed by something like a Bose-Einstein condensate. The model is used for investigating the relationship between galactic central black holes, dark matter and the plateau velocity of a galaxy rotation. Analytical solutions are supported by numerical results. The paper is a very well written and comprehensive review of recent interesting results results connecting fundamental questions of black hole physics. Its content is relevant and deserves further investigation. In special the change of metric signature between r_g and r_h.
According to the above comments, I recommend the manuscript for publication.
My only suggestion is to check the first sentence of the second line of page 21.
Author Response

(The authors gave the same response as above.)

Round 2
Reviewer 2 Report
This version of the manuscript presented by the author has improved greatly. The author has attended appropriately most of my initial concerns regarding this work. Therefore, in my opinion this document is now suitable for publication in Universe.
Reviewer 3 Report
I am convinced with the reply given by the author to my comments and modifications made by the author in the revised version. It seems the revised manuscript is suitable to be accepted for publication in the Universe journal.